# Melatonin-Mediated Regulation of Growth and Antioxidant Capacity in Salt-Tolerant Naked Oat under Salt Stress

**DOI:** 10.3390/ijms20051176

**Published:** 2019-03-07

**Authors:** Wenying Gao, Zheng Feng, Qingqing Bai, Jinjin He, Yingjuan Wang

**Affiliations:** Key Laboratory of Biotechnology of Shannxi Province, Key Laboratory of Resource Biology and Biothchnology in Western China (Ministry of Education), College of Life Science, Northwest University, Xi’an 710069, China; 201620849@stumail.nwu.edu.cn (W.G.); 201820916@stumail.nwu.edu.cn (Z.F.); 201731801@stumail.nwu.edu.cn (Q.B.); 201831860@stumail.nwu.edu.cn (J.H.)

**Keywords:** melatonin, naked oat, salt stress, plant growth, antioxidant capacity

## Abstract

Melatonin (MT; *N*-acetyl-5-methoxytryptamine) is a pleiotropic signaling molecule that has been demonstrated to play an important role in plant growth, development, and regulation of environmental stress responses. Studies have been conducted on the role of the exogenous application of MT in a few species, but the potential mechanisms of MT-mediated stress tolerance under salt stress are still largely unknown. In this study, naked oat seedlings under salt stress (150 mM NaCl) were pretreated with two different concentrations of MT (50 and 100 μM), and the effects of MT on the growth and antioxidant capacity of naked oat seedlings were analyzed to explore the regulatory effect of MT on salt tolerance. The results showed that pretreating with different concentrations of MT promoted the growth of seedlings in response to 150 mM NaCl. Different concentrations of MT reduced hydrogen peroxide, superoxide anion, and malondialdehyde contents. The exogenous application of MT also increased superoxide dismutase, peroxidase, catalase, and ascorbate peroxide activities. Chlorophyll content, leaf area, leaf volume, and proline increased in the leaves of naked oat seedlings under 150 mM NaCl stress. MT upregulated the expression levels of the lipid peroxidase genes *lipoxygenase* and *peroxygenase*, a chlorophyll biosynthase gene (*ChlG*), the mitogen-activated protein kinase genes *Asmap1* and *Aspk11*, and the transcription factor genes (except *DREB2*), *NAC*, *WRKY1*, *WRKY3*, and *MYB* in salt-exposed MT-pretreated seedlings when compared with seedlings exposed to salt stress alone. These results demonstrate an important role of MT in the relief of salt stress and, therefore, provide a reference for managing salinity in naked oat.

## 1. Introduction

Soil salinization is a global ecological problem and one of the main factors leading to land desertification and degradation of arable land. Salt stress in response to soil salinization has become an important factor limiting plant growth and yield [1,2]. In the long-term evolutionary process, plants have developed a series of mechanisms to maintain oxidative balance under various stressors [3]. When a plant is subjected to stress, the extracellular signals are amplified in the cell through a signal transduction pathway, such as mitogen-activated protein kinase (MAPK), which activates transcription factors (TFs) to bind downstream target genes and promotes the expression of related genes, thereby increasing the plants’ response to stress [4].

Salt stress induces the accumulation of reactive oxygen species (ROS) in plants, such as singlet oxygen (^1^O_2_^•−^), hydroxyl radical (OH^•^), superoxide anion (O_2_^•−^), and hydrogen peroxide (H_2_O_2_) [5], causing oxidative stress. Moreover, H_2_O_2_ can be used as a signaling molecule for the plant MAPK cascade to induce the expression of antioxidant-related genes, which maintain the balance between ROS production and clearance in plants [6]. MsERK1, which was discovered and identified in 1993, is the first MAPK found in plants [7]. MAPK genes were sequentially identified in *Arabidopsis thaliana*, rice, maize, wheat, and barley [8,9,10,11,12], and one study suggested that the MAPK cascade pathway is transmitted from environmental stress through MAPK-mediated phosphorylation cascades to enhance signals, such as high salinity, low temperature, and drought stress, which reach the nucleus and regulate downstream gene expression, thereby inducing the plant stress response [13]. Under high salt stress, *Arabidopsis* MPK6 activates the downstream TF MYB41, which improves salt tolerance in plants [14]. Low temperature stress activates ZmMPK17, which in turn activates downstream TFs to enhance the tolerance of maize to stress [15]. In addition, overexpressed GhMPK2 in transgenic cotton and MPK6 in *Arabidopsis* participate in plant stress tolerance in response to salt and drought stressors [16,17]. Therefore, the MAPK cascade pathway plays an important role in the response to abiotic stress.

Naked oat (*Avena nuda* L.) is an annual herbaceous crop in the family *Poaceae*. It is a unique food crop found in Chinese alpine regions. It has many physiological functions, such as lowering blood sugar, lowering blood fat, and scavenging free radicals [18,19]. In recent years, naked oat has been the focus of attention by researchers with application prospects due to its extremely high nutritional value and healthcare functions. However, abiotic stressors, such as drought and salt, directly limit the growth of naked oat, restricting effective utilization and development of the plant [20].

Melatonin (MT; *N*-acetyl-5-methoxytryptophan) was first found in the pineal gland of cattle [21]. Thereafter, further studies showed that MT occurs in mammals, birds, amphibians, and fish where it acts as a hormone involved in many physiological processes, including circadian rhythms, mood, temperature homeostasis, sleep physiology, spontaneous activity, food intake, and retinal physiology [22,23,24]. MT was found in many higher plants in the 1990s [20], and an increasing number of studies have revealed the important role of exogenous application of MT during all stages of plant growth. MT regulates root and stem growth [25], seed germination [26], cell division [27], crop yield [28] and alleviates oxidative damage [29,30]. In addition, MT protects plants against multiple abiotic stressors, such as extreme temperature conditions [31], heavy metal stress [32], and salt stress [33]. The application of MT in stress conditions is attributable to higher photosynthesis, scavenger of ROS in cell, and regulation of the expression of stress-responsive genes involved in signal transduction [29,30]. Recently, several studies have shown that MT confers salt stress tolerance by improving photosynthesis and redox homeostasis in watermelon [30] and exogenous application of MT enhances salt stress tolerance in cucumber [34] and *Malus hupehensis* [35]. Therefore, in the present study, we elucidated the regulatory mechanism controlling MT-mediated salt stress tolerance in naked oat. We hypothesized thatexogenous application of MT would enhance the antioxidant capacity of naked oat seedlings under NaCl stress and thereby enhancing the antioxidant capacity of naked oat seedlings under NaCl stress. 

## 2. Results

### 2.1. Effect of Melatonin (MT) Pretreatment on the Growth of Naked Oat Seedlings under Salt Stress

The salt treatment resulted in decreased plant height, stem thickness, plant fresh weight, and plant dry weight. However, pretreatment with 50 or 100 μM MT alleviated the salt stress-induced reductions in these indicators, and 100 μM MT appeared to be much more effective for alleviating NaCl stress (Figure 1). For example, salt stressed plants revealed increased height, stem thickness, fresh weight, and dry weight by 23.71%, 48.00%, 21.43%, and 35.62%, respectively, after pretreatment with 100 μM MT, but only by 14.10%, 24.00%, 11.56%, and 23.29%, respectively, after pretreatment with 50 μM MT. These results indicate that the exogenous application of MT significantly attenuated the inhibitory effect of NaCl stress on the growth of plant seedlings. 

The leaves of the salt-treated seedlings turned yellow after six days of treatment, indicating salt toxicity symptoms (Figure 2). Treating the seedlings with exogenous application of MT under salt stress alleviated the symptoms of salt toxicity, which was manifested as greening of the leaves and the growth of plants (Figure 2 and Figure 3). Compared with the salt group, chlorophyll content, leaf area, and leaf volume of naked oat averagely increased by 31.25%, 11.56%, and 36.21%, respectively, after NaCl stress with MT pretreatment. In addition, the expression level of *ChlG*, which is a chlorophyll biosynthesis gene, was upregulated 1.69-fold (Figure 3). These results demonstrate that application of MT may enhance naked oat seedlings tolerance to salt stress by improving photosynthesis.

### 2.2. Effect of MT Pretreatment on Antioxidant Enzyme Activities in Naked Oat Seedlings under Salt Stress 

Antioxidant enzyme activities remained almost unchanged under normal hydroponic conditions after pretreatment with different MT concentrations, except for the significant (*p* < 0.01) increase in SOD activity compared with the control group (Figure 4). In contrast, the activities of SOD, POD, CAT, and APX increased in leaves after the plants were exposed to 150 mM NaCl for six days. POD and APX activities were significantly (*p* < 0.01) higher in the plants pretreated with 100 μM MT than those in control plants after 150 mM NaCl stress. Moreover, higher CAT activity was detected in plants pretreated with 50 μM MT after salt stress. SOD activity increased significantly (*p* < 0.01) after salt stress, despite the 50 or100 μM MT pretreatment. These results indicate that exogenous application of MT enhanced antioxidant enzyme activities to scavenge ROS under salt stress, thereby promoting growth of the naked oat seedlings.

### 2.3. Effect of MT Pretreatment on the Changes in Osmotic Adjusting Substances in Plant Seedlings under Salt Stress

Soluble protein content increased slightly after pretreatment with 50 or 100 μM MT compared with the control group (Figure 5A). However, the NaCl-induced decrease in soluble protein content was alleviated by pretreatment with 50 or 100 μM MT. Soluble protein contents were 141.24% and 74.23% higher than those in control plants after the NaCl treatment. Proline increased significantly after six days of exposure to NaCl (Figure 5B) compared with the control group, and proline content was higher under the NaCl treatment combined with the MT pretreatment in naked oat. The MT pretreatment increased proline content of naked oat seedlings in a dose-dependent manner (Figure 5B). These data suggest that osmotic adjustment regulate cell osmotic pressure and maintain the metabolic system, thereby enhancing salt tolerance of the naked oat seedlings.

### 2.4. Effect of MT Pretreatment on Changes in Reactive Oxygen Species (ROS) in Naked Oat Seedlings under Salt Stress 

The H_2_O_2_ and O_2_^•−^ contents in naked oat seedlings increased in response to salt stress. However, the NaCl-induced increases in H_2_O_2_ and O_2_^•–^ were attenuated by pretreatment with 50 or 100 μM MT (Figure 6A,B). Similarly, MDA, which reflects cell membrane damage, increased significantly in response to NaCl stress in control plants, but this increase was attenuated by the exogenous application of MT pretreatment (Figure 6C). For example, the average H_2_O_2_, O_2_^•−^, and MDA contents in the naked oat seedlings decreased significantly (*p* < 0.01) by 33.51%, 26.25%, and 50.50%, respectively, under salt stress after the MT pretreatment, compared with the salt group. These results indicate that MT can scavenge excessive accumulation of ROS and reduce lipid peroxidation, thereby improving salt stress tolerance in naked oat seedlings.

### 2.5. Effect of MT Pretreatment on Intracellular Lipid Peroxidase Genes in Naked Oat Seedlings under Salt Stress

Lipoxygenase (LOX) plays an important role in plant resistance to multiple abiotic stressors [36]. In this study, the expression of the *LOX* and *peroxygenase (POX*) in naked oat leaves was not different from that in the control group after the 50 or 100 μM MT pretreatment (Figure 7). However, the expression levels of both the *LOX* and *POX* were upregulated after NaCl stress in control plants, and this upregulation was higher after the pretreatment with 100 μM MT (Figure 7). The relative expression levels of the two genes were 2.88-fold and 2.63-fold higher than those in the salt stress alone group of seedlings. *LOX* expression was significantly upregulated (*p* < 0.05) after the NaCl treatment and 50 μM MT pretreatment, compared to salt stress alone (Figure 7A), whereas expression of the *POX* appeared to be less affected under salt stress in the groups of naked oat seedlings with or without the MT pretreatment (Figure 7B). These data show that *LOX* and *POX* functions positively in the plant response to the salt stress.

### 2.6. Effect of MT Pretreatment on the Expression of Mitogen-Activated Protein Kinase (MAPK) Genes in Naked Oat Seedlings under Salt Stress

No significant difference was observed for both *Asmap1* and *Aspk11* after the MT pretreatment, compared to the control plants (Figure 8). The NaCl stress inhibited the growth of naked oat, while the MT pretreatment alleviated this inhibition to a certain extent (Figure 8). Expression of the *Aspk11* was higher in 100 μM MT-pretreated plants than that in 50 μM-pretreated MT plants after the NaCl stress (Figure 8B). However, *Asmap1* expression was significantly (*p* < 0.01) upregulated only in the 50 μM MT-pretreated plants (Figure 8A). Interestingly, the relative expression of the *Asmap1* was lower after NaCl stress and the 100 μM MT pretreatment than that of the salt stress group (*p* < 0.05) (Figure 8A). These results demonstrate that MT might induce a MAPK cascade through the H_2_O_2_ pathway, which regulates the expression of MAPK genes, thereby enhancing naked oat stress tolerance. Further research is required to charity the exact mechanism by which MT affects MAPK pathway.

### 2.7. Effect of MT Pretreatment on the Expression of Transcription Factors (TFs) in Naked Oat Seedlings under Salt Stress

Salt stress affected the expression of antioxidant-related TF genes in naked oat seedlings (Figure 9). The MT pretreatment increased the expression levels of the *NAC*, *WRKY3,* and *MYB* in naked oat seedlings at the transcriptional level compared with the control group (Figure 9A,C,D), but the expression level of the *WRKY1* was not different between plants treated with the two MT concentrations (Figure 9B). The expression level of *DREB2* decreased significantly (*p* < 0.01) (Figure 9 E) after the MT pretreatment. Similarly, after six days of the salt treatment, our data confirmed that the salt condition upregulated the expression levels of *NAC*, *WRKY1*, *WRKY3*, *MYB*, and *DREB2* in naked oat seedlings (Figure 9), and a pre-application of 50 or 100 μM MT to the plants under NaCl stress also promoted that upregulation. For example, the average expression levels of *NAC*, *WRKY1*, *WRKY3*, and *MYB* in the plants pretreated with 50 or 100 μM MT followed by salt stress were upregulated 2.72-fold, 8.80-fold, 1.14-fold, and 0.47-fold, respectively, than those in the control plants after NaCl stress (Figure 9 A–D). However, the 50 or 100 μM MT pretreatment significantly (*p* < 0.01) inhibited expression of the *DREB2* in plant seedlings under salt stress (Figure 9E). These results demonstrate that MT might induce a MAPK cascade through the H_2_O_2_ pathway, which regulates the expression of TF genes, thereby enhancing naked oat stress tolerance. Further research is required to clarify the exact mechanism by which this is achieved.

## 3. Discussion

In recent years, functional research on MT has become important. MT has various biological functions in plants, such as morphogenesis of organs [30], protection of chlorophyll [37], delay of leaf senescence [38], and promotion of growth [39]. In the present study, NaCl stress inhibited plant growth and biomass accumulation in naked oat. However, pretreatment with 50 or 100 μM MT alleviated the salt-induced inhibition of plant growth and biomass accumulation, which agrees with earlier studies on *Malus hupehensis* [35] and maize [40]. Chlorophyll content and the expression level of *ChlG* in seedlings decreased under salt stress (Figure 3), while pretreatment with 50 or 100 μM MT alleviated the salt-induced inhibition in chlorophyll content and *ChlG* expression (Figure 1 and Figure 3). Protection of chlorophyll has also been observed in cucumber [39] and maize [40] under a salt stress condition.

ROS and metabolism increase in salt-stressed plants, and cell membrane functions change or are destroyed. Since the biological free radical damage theory was proposed by Fridovich [41] in 1975, most scholars have explored the defense mechanisms of plants under adverse conditions from the perspective of antioxidant enzymes and the membrane lipid peroxidation product MDA, which are related to the production and clearance of ROS in plants. In the current study, the activity of antioxidant enzymes (SOD, POD, CAT, and APX) increased in MT-pretreated naked oat, which was beneficial to the growth of naked oat under salt stress. Similar studies have been reported in maize [40], bermudagrass [42], and watermelon [30]. At the same time, the salt treatment increased H_2_O_2_, O^2−^• O_2_^•–^, and MDA contents in the leaves of naked oat seedlings, while the MT pretreatment reduced H_2_O_2_, O_2_^•–^, and MDA contents (Figure 4), effectively inhibiting the increase in ROS and MDA in leaves of naked oat seedlings under salt stress. These results were similar to the effect of MT on watermelon [35] and *Chlamydomonas reinhardtii* [43] and show that MT induced an increase in antioxidant enzyme activities in naked oat leaves to remove excess ROS and protect plants against oxidative stress. 

ROS are aerobic cellular metabolites that damage cells under stress conditions. H_2_O_2_ is an important ROS signaling molecule in the plants’ response to stress, and it is widely involved in plant physiology and cross-resistance processes [44]. H_2_O_2_ acts as a second messenger in the signal transduction of brassinosteroids (BRs) and abscisic acid (ABA)-induced plant resistance [45]. Xia et al. [45] speculated that BRs might induce a MAPK cascade through the H_2_O_2_ pathway, which regulates the expression of antioxidant enzyme genes, thereby promoting the activity of antioxidant enzymes, which enhance cucumber stress tolerance. MT had a similar effect in the present study. MT might induce a MAPK cascade through a H_2_O_2_ signaling pathway to control the expression of genes involved in salt stress-related signal transduction, thereby increasing the plant’s tolerance. The 50 or 100 μM MT pretreatment increased antioxidant enzyme activities and upregulated the expression of MAPK and TF genes (Figure 4, Figure 8, and Figure 9). These results further confirmed the two effects of MT: (i) MT scavenges ROS by enhancing antioxidant enzyme activities; (ii) MT induces a MAPK cascade through the H_2_O_2_ pathway, thus, improving salt tolerance of naked oat seedlings. These results help clarify the underlying mechanism of MT.

Salt has an adverse effect on plant metabolism through osmotic and oxidative stress [46]. The plant osmotic pressure reaction mainly involves osmotic adjustment to regulate cell osmotic pressure and maintain the metabolic system, which affects plant growth and yield [47]. During osmotic adjustment, plants synthesize soluble protein, proline, and other permeates to complete the cell-level balance of penetration [48]. In the present study, salt stress caused hypoxia, resulting in a large quantity of peroxidation products in the plant, causing damage to soluble proteins, resulting in a decrease in contents. However, pretreatment with MT after salt stress increased soluble protein contents possibly because MT acts as a ROS scavenger to reduce excess intracellular ROS and repair protein damage under salt stress. Proline scavenges ROS, improves antioxidant capacity, stabilizes biomacromolecular structure, reduces cell acidity, relieves ammonia toxicity [49,50], and plays an important role in plant osmotic regulation. Our data show that proline content in the leaves of naked oat seedlings increased significantly during salt stress after pretreatment with MT. It was opposite to the changing trend in soluble protein content, which better repairs the cell damage of plants in an adverse environment and is conducive to plant growth. Therefore, proline may be the main organic osmotic adjustment substance in naked oat seedlings under NaCl stress.

LOX catalysis produces membrane lipid peroxidation substances, such as hydroxides and oxygen-containing free radicals, which participate in environmental stress through signal transduction pathways [51]. A variety of LOX genes have been cloned from various plants and are closely related to the stress response. For example, overexpressing the *TOMLOXC* in tomato enhances plant resistance through the accumulation of jasmonic acid (JA) [52]; *Arabidopsis* transgenic plants overexpressing the *CaLOX1* in pepper leaves have significantly enhanced resistance though ABA, drought, and salt [53]. In the present study, salt stress also increased expression of the *LOX* in the leaves of naked oat seedlings, and the expression of *LOX* was higher after pretreatment with exogenous application of MT. This result indicates that salt stress induces expression of the *LOX* and supports the fact that LOX enhances the abiotic stress resistance of plants [53] in a variety of ways.

The MAPK cascade responds to various biotic and abiotic stressors in plants, such as bacteria, high salinity, drought and oxidative stress [16,54,55]. Phosphorylated MAPK acts as a bridge between upstream receptors and downstream TF genes and stimulates TFs to regulate the expression of downstream resistance genes, thereby enhancing plant resistance [56]. The TFs NAC, WRKY, MYB, and DREB, which are closely related to the plant stress response, regulate against adverse living environments [57]. In a similar study, flagellin flg22 triggered a complete MAPK cascade, and MEKK1-MKK4/5-MAPK3/6 sequentially activated expression of the downstream gene *WRKY22/29,* thereby enhancing *Arabidopsis* resistance to bacterial and fungal pathogens [58]. In addition, the WRKY TF family (OsWRKY45 and OsWRKY72) in *A. thaliana* enhances the expression of downstream genes through the ABA signaling pathway and participates in improving salt tolerance of *A. thaliana* [59]; In this study, the relative expression levels of two MAPK protein genes, such as *Asmap1* and *Aspk11*, were significantly upregulated under salt stress (*p* < 0.05). If MT was applied as a pretreatment, the relative expression of both genes increased or decreased in plant leaves depending on the MT concentration (Figure 8). The expression levels of the TFs *NAC*, *WRKY1*, *WRKY3,* and *MYB* were upregulated under salt stress. When treated with different concentrations of MT, the expression levels of these genes were significantly higher (*p* < 0.01) than those of plants exposed to salt stress alone (Figure 9A–D). Interestingly, the expression of *DREB2* increased, while the expression of *DREB2* decreased significantly (*p* < 0.01) under salt stress with the MT treatment (Figure 9E). Therefore, NAC, WRKY1, WRKY3, MYB, and DREB2 in the TF family may be involved in the MAPK-related signaling pathway regulated by ROS in naked oat.

In conclusion, the potential mechanism of MT-mediated enhancement of salt tolerance in naked oat seedlings is shown in Figure 10. MT promoted the production and accumulation of ROS in cells. As the concentration of ROS increased, MT acted as an antioxidant to scavenge ROS, thereby enhancing the activity of antioxidant enzymes; H_2_O_2_ among ROS increased the expression of antioxidant enzyme genes/MT biosynthesis-related genes, and acted as a signaling molecule to phosphorylate MAPKs. The phosphorylated MAPKs promoted the expression of TFs genes, such as *NAC*, *WRKY1*, *WRKY3*, *MYB*, and *DREB2*, and then regulated the expression of downstream resistance genes, thereby enhancing salt tolerance in naked oat. The MT-mediated signaling pathways related to the salt stress response are complex, and further research is needed.

## 4. Materials and Methods

### 4.1. Plant Materials

Naked oat seeds of the ‘Jin Yan No. 2’ cultivar (*A. nuda* L.) were provided by the Provincial Key Laboratory of Biotechnology of Shaanxi Province, Northwest University, Xi’an 710069, Shannxi Province, China. They were stored at 4 °C under dry conditions, before the experiments started. 

### 4.2. Seedlings Growth Conditions

Naked oat seeds were surface-sterilized with a 75% (*v/v*) alcohol solution for 15 s, washed two to three times in distilled water, sterilized with 1% HgCl_2_ (*m/v*) for seven min, and washed again five to six times in distilled water. The seeds were soaked in a 100 μM concentration for 12 h, then placed on 14-cm diameter Petri dishes (40 seeds per dish) with three layers of filter paper in a dark growth chamber (23–25 °C/16–18 °C day/night) for germination. After emergence of the radicle, naked oat seedlings were transferred to Hoagland nutrient solution for hydroponic culture (pH 6.5 ± 0.1). The naked oat seedlings were placed in a culture room with a relative humidity of 75%, day/night temperatures of 25 °C/17 °C, and a 14L/10D photoperiod.

### 4.3. MT Pretreatment 

After five days, two-thirds of the seedlings were transferred to half-strength Hoagland solution containing 50 or 100 μM MT, while the remaining seedlings were transferred to half-strength Hoagland solution. MT (50 or 100 μM) was applied to the naked oat seedlings for the first time at eight p.m. and the plants were treated once every day. Exogenous application of 50 or 100 μM MT was treated three times over an experimental period of six days. On the basis of the naked oat growth in our present study [5], we selected two different concentrations of MT (50 or 100 μM). 

### 4.4. Salt-Stress Pretreatment 

After two days of MT pretreatment, half of the MT-pretreated and non-MT-pretreated seedlings were transferred to half-strength Hoagland solution while the other half were transferred to half-strength Hoagland solution containing 150 mM NaCl for six days. The leaves were harvested, rapidly frozen in liquid nitrogen, and stored at −80 °C for further analysis. According to the growth states of the naked oat seedlings under different salt stress conditions (Appendix A), the concentration of 150 mM NaCl was selected based on our result. In the measurement process, each treatment group contained 90 strains of seedlings (each contained 30 strains in three parallel control groups), and randomly selected six strains of 90 seedlings in three replicates for various indicators, and each index was repeated in the biological and experimental for three times (that is, in the biological and experimental replicates for three times, 18-strain seedlings were randomly selected from 270 seedlings).

### 4.5. Calculation of Plant Height, Stem Thickness, Plant Fresh Weight, and Plant Dry Weight 

After treatment in 150 mM NaCl for six days, seedlings were randomly selected from each dish and plant height, stem thickness, plant fresh weight, and plant dry weight were measured. The plant height was the linear distance from the base of the radicle to the top of the blade. Plants were harvested, then washed with tap water and rinsed for two~three times with distilled water, gently wiped dry with a paper towel and their fresh weight was determined rapidly. Then, the seedlings were dried at 105 °C for 30 min, and dried at 80 °C for 24 h to measure their dry weight. All the experiments were performed for three times.

### 4.6. Calculation of Chlorophyll Content, Leaf Area, and Leaf Volume

After the six-day salt treatment, chlorophyll content, leaf area, and leaf volume of the seedlings were calculated. Leaf area was calculated using the formula: leaf area = leaf length × maximum blade width; chlorophyll volume calculation formula: (C_v_) = C_t_ −10/T. C_t_ was the total volume of leaves with 10 mL of distilled water, and T was the total number of naked oat seedlings in the calculation formula. Leaf samples (100 mg) were extracted in 60% acetone and the contents of chlorophyll a and chlorophyll b were determined according to the method of the manufacturer’s instructions (Nanjing Institute of Bioengineering, Nanjing, China). The reaction mixtures were then incubated at 4 °C and the absorbance was recorded at 645 nm and 663 nm with a spectrophotometer.

### 4.7. Enzyme Extraction and Assay

Antioxidant enzyme activities were assayed in leaves by using kit methods. All enzymes were extracted by grinding 100 mg of fresh leaves with 900 μL of ice-cold 100 mM phosphate buffer (pH 7.4) or saline-containing liquid nitrogen using a chilled mortar and pestle. The homogenate was centrifuged at 3500 rpm/min for 10 min at 25 °C, and the supernatant was used for the specific enzyme activity assays. The activities of superoxide dismutase (SOD; EC 1.15.1.1), peroxidase (POD; EC 1.11.1.7), catalase (CAT; EC 1.11.1.6) and ascorbate peroxide (APX; EC 1.11.1.11) were assayed by the method described by Zhang et al [43]. SOD, POD, CAT, and activities were recorded at a wavelength of 550 nm, 420 nm, 405 nm, and 290 nm, respectively. 

### 4.8. Determination of Soluble Protein and Proline Contents 

Soluble protein and proline contents were determined according to the instructions described in soluble protein and proline kits (Nanjing Institute of Bioengineering, Nanjing, China). 100 mg of fresh leaves was homogenized 900 μL buffer and the homogenate was centrifuged at 3000 rpm/min for 10 min; 50 μL supernatant was added to 3 mL coomassie brilliant blue solution. The mixture was incubated at 25 °C for 30 min and the absorbance of soluble protein content was recorded at 595 nm; 50 μL supernatant was added to 2 mL reagent solution. The mixture was incubated in a boiling water bath for 30 min and terminated in an ice bath and an absorbance of proline content was recorded at 520 nm.

### 4.9. Determination of H_2_O_2_, O_2_^•–^ and Malondialdehyde (MDA) 

To measure H_2_O_2_ and MDA, fresh leaves (100 mg) were ground in a mortar with 900 μL buffer, following the methods described by Nawaz et al [60]. H_2_O_2_ and MDA contents were recorded at a wavelength of 405 nm and 532 nm, respectively; O_2_^•−^ was evaluated according to the instructions described in O_2_^•–^ kit purchased from Nanjing Jiancheng Bioengineering Institute, Nanjing, China. Fresh leaves (100 mg) were homogenized in 900 μL buffer and centrifuged at 4000 rpm/min and 25 °C for 10 min. 50 μL of the supernatant was mixed with the 4 mL reagent solution. The mixture was bathed at 37 °C for 40 min and then 2 mL reagent solution was added. O_2_^•–^ content was recorded at a wavelength of 550 nm.

### 4.10. RNA Extraction and Gene Expression Analysis

The total RNA of naked oat leaves was extracted by TRIzol reagent (Invitrogen, Carlsbad, CA, USA) method. First-strand cDNA was synthesized using the PrimeScript^TM^ RT reagent kit with the gDNA Eraser (Takara, Shiga, Japan) according to the manufacturer’s instructions. QRT-PCR was performed on a Bio-Rad CFX96 Real-Time PCR System (Bio-Rad, Hercules, CA, USA) using FastStart Essential DNA Green Master (Tiangen, Beijing, China). The procedure was performed as described previously [5]. The expression of the genes was calculated by the method described by Gao et al. [5] and normalized to *Actin* (KP257585.1). The list of primers is shown in Table 1.

### 4.11. Statistical Analysis 

The experiments were divided into a control untreated group (Con: seedlings grown in half-strength Hoagland solution), different MT concentration treatment groups (MT_5_ and MT_10_: seedlings pretreated with 50 and 100μM MT, grown in half-strength Hoagland solution), the salt stress treatment group (S: seedlings grown in half-strength Hoagland solution plus 150mM NaCl) and the salt stress treatment and different MT concentration treatment groups (S_5_ and S_10_: seedlings pretreated with 50 and 100 μM MT, grown in half-strength Hoagland solution plus 150 mM NaCl). All experiments were repeated three times. Analysis of variance and Student’s *t*-test were used to test for differences compared with the mock treatment. A *p*-value < 0.05 was considered significant. 

## Figures and Tables

**Figure 1 ijms-20-01176-f001:**
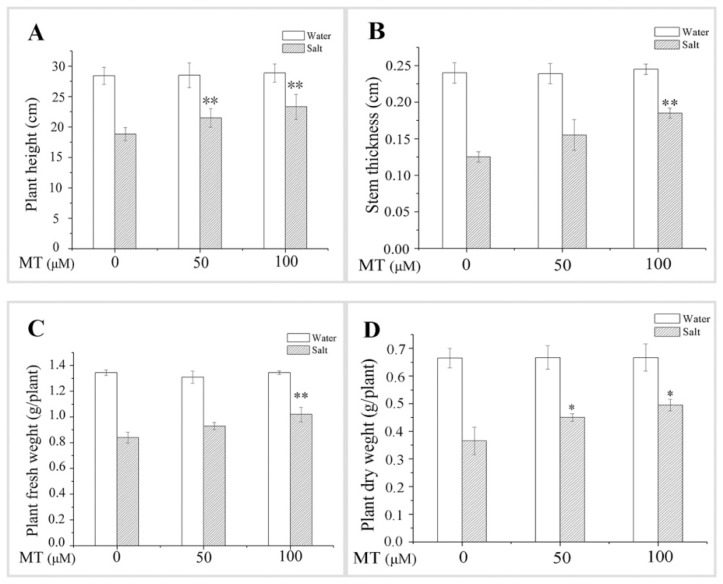
Effect of pretreating with different concentrations of melatonin (MT) on growth of leaves of naked oat seedlings under salt stress. (**A**) Plant height; (**B**) stem thickness; (**C**) plant fresh weight; (**D**) plant dry weight. Values are mean ± standard deviation (*n* = 3). 0, 50, and 100 represent MT concentration (μM) in Figure 1. Asterisks; * and ** indicate significant difference (*p* < 0.05 and *p* < 0.01, respectively) compared with mock coating (0 μM). Measurement was carried out after six days of the salt plus MT pretreatmemt.

**Figure 2 ijms-20-01176-f002:**
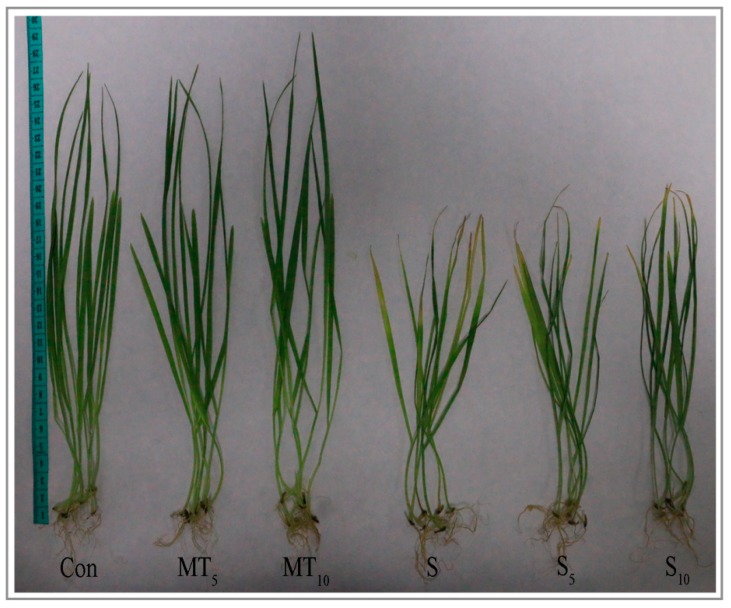
The performance of plant seedlings under normal and salt stress conditions. Photographs were taken after six days of the salt plus melatonin (MT) pretreatment.

**Figure 3 ijms-20-01176-f003:**
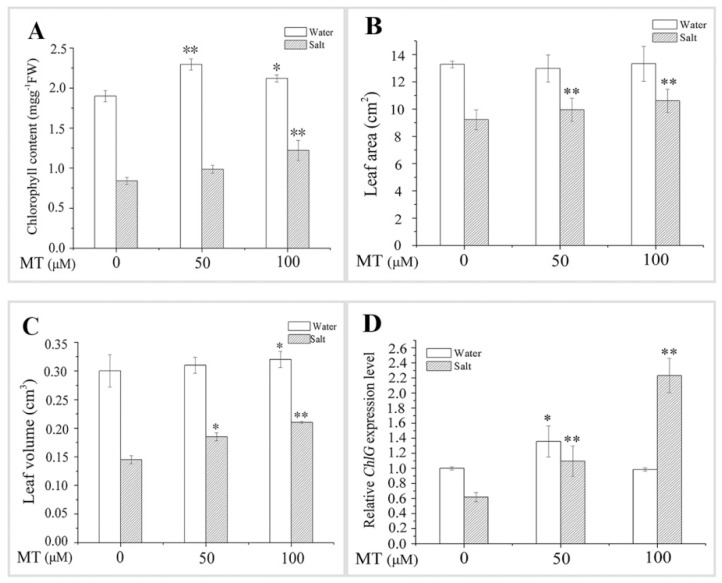
Effect of pretreating with different concentrations of melatonin (MT) on growth of leaves of naked oat seedlings under salt stress. (**A**) Chlorophyll content; (**B**) leaf area; (**C**) leaf volume; (**D**) relative chlorophyll biosynthase gene (*ChlG*) expression level. Values are mean ± standard deviation (*n* = 3). 0, 50, and 100 represent MT concentration (μM) in Figure 3. Asterisks; * and ** indicate significant difference (*p* < 0.05 and *p* < 0.01, respectively) compared with mock coating (0 μM). Measurement was carried out after six days of the salt plus MT pretreatmemt.

**Figure 4 ijms-20-01176-f004:**
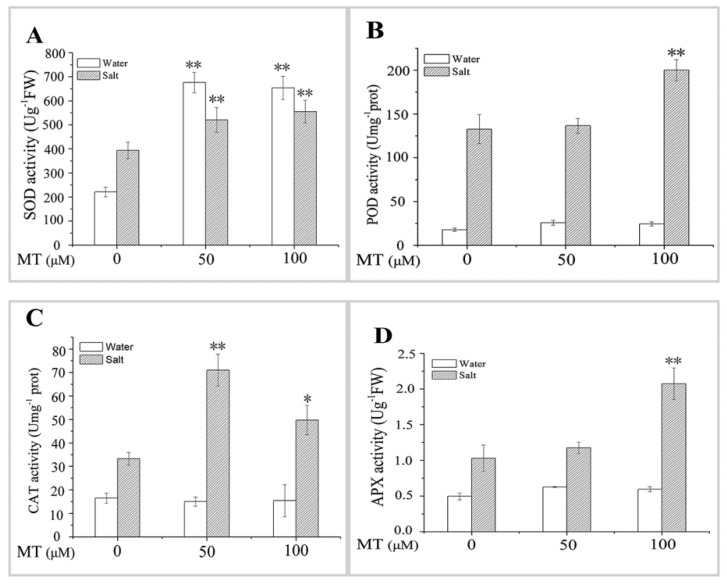
Effect of pretreating with different concentrations of melatonin (MT) on the antioxidant enzyme activities of leaves of naked oat seedlings under salt stress. (**A**) Super oxide dismutase (SOD); (**B**) peroxidase (POD); (**C**) catalase (CAT); (**D**) ascorbate peroxidase (APX). Values are mean ± standard deviation (*n* = 3). 0, 50, and 100 represent MT concentration (μM) in Figure 4. Asterisks; * and ** indicate significant difference (*p* < 0.05 and *p* < 0.01) compared with mock coating (0 μM). Measurement was carried out after six days of the salt plus MT pretreatmemt.

**Figure 5 ijms-20-01176-f005:**
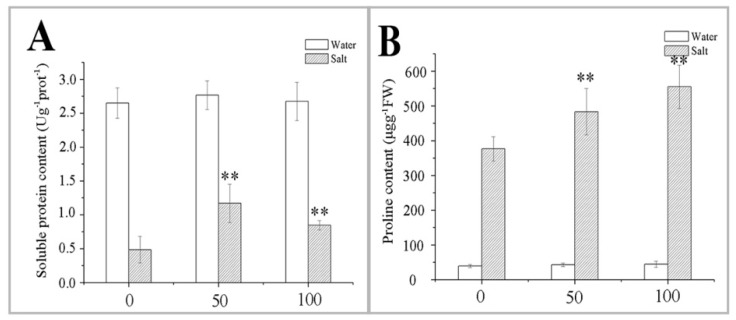
Effect of pretreating with different concentrations of melatonin (MT) on the changes of osmotic adjusting substances in leaves of naked oat seedlings under salt stress. Soluble protein content (**A**); and proline content (**B**).Values are mean ± standard deviation (*n* = 3). 0, 50, and 100 represent MT concentration (μM) in Figure 5. Asterisks; * and ** indicate significant difference (*p* < 0.05 and *p* < 0.01) compared with mock coating (0 μM). Measurement was carried out after six days of the salt plus MT pretreatmemt.

**Figure 6 ijms-20-01176-f006:**
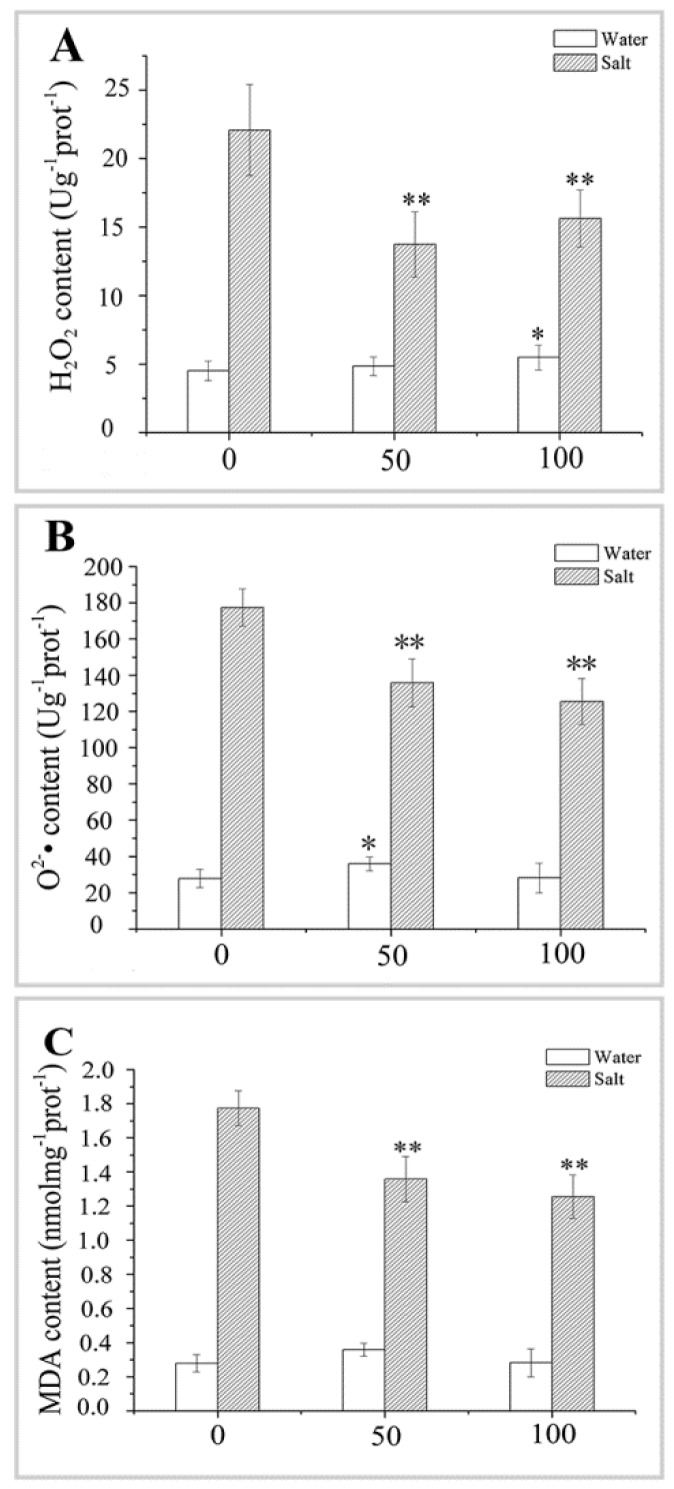
Effect of pretreating with different concentrations of melatonin (MT) on the changes of ROS in leaves of naked oat seedlings under salt stress. (**A**) Hydrogen peroxide (H_2_O_2_) content; (**B**) superoxide anion (O_2_^•–^) content; and (**C**) malondialdehyde (MDA) content. Values are mean ± standard deviation (*n* = 3). 0, 50, and 100 represent MT concentration (μM) in Figure 6. Asterisks; * and ** indicate significant difference (*p* < 0.05 and *p* < 0.01) compared with mock coating (0 μM). Measurement was carried out after six days of the salt plus MT pretreatmemt.

**Figure 7 ijms-20-01176-f007:**
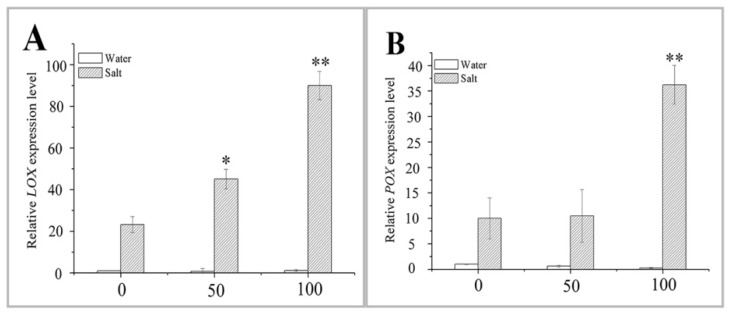
Effect of pretreating with different concentrations of melatonin (MT) on the expression of *lipoxygenase* (*LOX*) and *peroxygenase* (*POX*) in leaves of naked oat seedlings under salt stress. The relative expression of genes *LOX* (**A**) and *POX* (**B**). Values are mean ± standard deviation (*n* = 3). 0, 50, and 100 represent MT concentration (μM) in Figure 7. Asterisks; * and ** indicate significant difference (*p* < 0.05 and *p* < 0.01) compared with mock coating (0 μM). Measurement was carried out after six days of the salt plus MT pretreatmemt.

**Figure 8 ijms-20-01176-f008:**
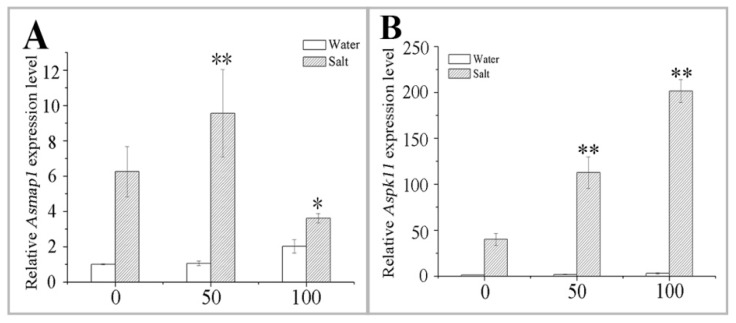
Effect of pretreating with different concentrations of melatonin (MT) on the expression of *MAPK* in leaves of naked oat seedlings under salt stress. The relative expression of MAPK genes *Asmap1* (**A**) and *Aspk11* (**B**). Values are mean ± standard deviation (*n* = 3). 0, 50, and 100 represent MT concentration (μM) in Figure 8. Asterisks; * and ** indicate significant difference (*p* < 0.05 and *p* < 0.01) compared with mock coating (0 μM). Measurement was carried out after six days of the salt plus MT pretreatmemt.

**Figure 9 ijms-20-01176-f009:**
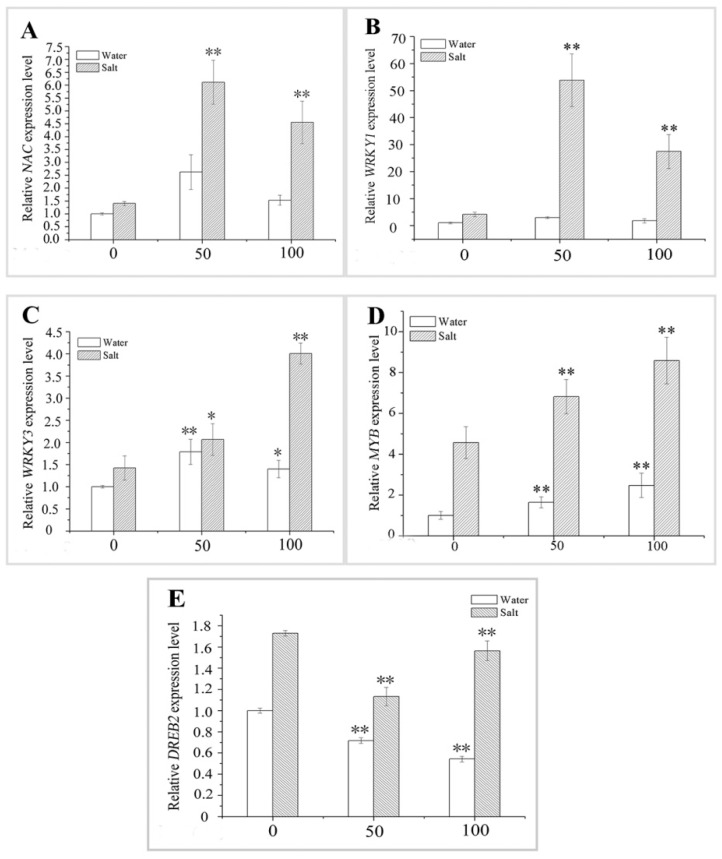
Effect of pretreating with different concentrations of melatonin (MT) on the expression of antioxidant-related transcription factors (TFs) genes in leaves of naked oat seedlings under salt stress. The relative expression of the related TFs genes *NAC* (**A**); *WRKY1* (**B**); *WRKY3* (**C**); *MYB* (**D**); and *DREB2* (**E**). Values are mean ± standard deviation (*n* = 3). 0, 50, and 100 represent MT concentration (μM) in Figure 9. Asterisks; * and ** indicate significant difference (*p* < 0.05 and *p* < 0.01) compared with mock coating (0 μM). Measurement was carried out after six days of the salt plus MT pretreatmemt.

**Figure 10 ijms-20-01176-f010:**
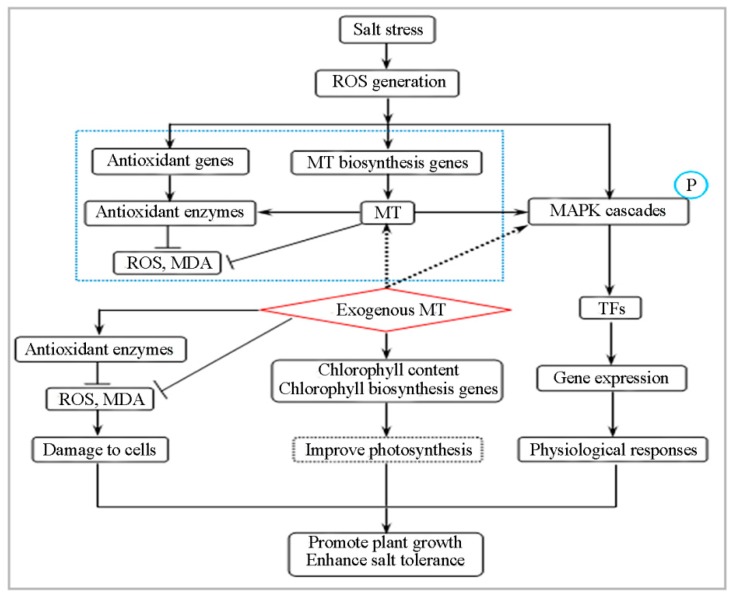
A model showing the potential mechanisms of melatonin (MT)-mediated alleviation in NaCl-caused oxidative stress of naked oat.

**Table 1 ijms-20-01176-t001:** Salt-tolerance relative genes and primers used in quantitative real-time RT-PCR analysis.

Primer	Accession No.	Sequence (5’–3’)
*ChlG*–Fq	AJ277210.1	CTTCCTGTTGCTTTTGGT
*ChlG*–Rq		GCTCGCCTGGTATTTGAC
*LOX*–Fq	JN390967.1	TCAACAACCTGGACGGCAACTTC
*LOX*–Rq		CGGCGTGTAGACCTTGCTCTTG
*POX*–Fq	JN390966.1	TGCAGCAGCATGTGTCCTTCTTC
POX–Rq		ATGGTCTCCAGGTAGGCAGAGTTG
*Asmap1*–Fq	X79993.1	CATCCGCTCCAACCAAGAACTCTC
*Asmap1*–Rq		TACTCCGTCATCATGTCGCTCTCC
*Aspk11*–Fq	X79992.1	GGTCCATACCCCCACAGA
*Aspk11*–Rq		TAGTCCAACAGCCCTCATT
*NAC*–Fq	KU886332.1	GGAGTCGGAGATCGTGGACACC
*NAC*–Rq		TGGATGTCGTCGTAGCTGAGGTC
*WRKY1*–Fq	AF140554.1	GGCGTCCTCCTTCCTCCAGTC
*WRKY1*–Rq		CCTCGTATGGCGTGCTGAAGC
*WRKY3*–Fq	AF140553.1	GACAGCAGCAGCAGCAGCAG
*WRKY3*–Rq		ACGAAGACGCCGTCCTCACC
*MYB*–Fq	AJ133638.1	GAACCAGCAGCCGTCTGTGAG
*MYB*–Rq		GCAGGAGCGGTGGATTCAGTG
*DREB2*–Fq	EF672101.1	ATACCGTGGTGTGAGGCAG
*DREB2*–Rq		CGAGATACGAGAAGGAGGA
*Actin*–Fq	KP257585.1	ATGTTGCCATCCAGGCTGTG
*Actin*–Rq		TAAGTCACGTCCAGCGAGGT

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
