# Peer review of "Melatonin-Mediated Regulation of Growth and Antioxidant Capacity in Salt-Tolerant Naked Oat under Salt Stress"

_ijms, 2019, doi:10.3390/ijms20051176_

Round 1
Reviewer 1 Report
I have added my comments on the PDF file of the manuscript.

Reviewer 2 Report
There are some issues in this manuscript for publication.
Major concerns
Most of all, my major concern in this manuscript is that authors used low number of samples in all physiological results. These low number of samples did not seem to show statistical meaning. As shown in Figure 1, I could find several number of leaves even in one seedling. What does 'n=3' mean? Authors must clarify the number of samples, how you selected samples and how you used them in measurement in figure legend or materials and methods. Or authors seem to need to use an increased number of samples in physiological experiments.
Minor concerns
1) In line 51 and 54, authors need to correct gene names, such as MAPK6 (line 51) and MAK6 (line 54), as “MPK6” based on references what authors provided.
2) In line 101, authors need to change unclear sentence, “After 6 days of the salt plus MT pretreatment” like as “Measurement was carried out after 6 days of the salt plus MT pretreatment” in each figure legend.
3) In line 126-128, authors described the conclusion of the biological role of MT to enhance antioxidant enzyme activities. However, authors showed the attenuated result of SOD in Figure 3A, whereas other enzymes displayed the opposite roles in the condition containing MT. Is there any different effect of SOD compared to other antioxidant enzymes including POD, CAT and APX? Authors need to add more discussion for this or change the conclusion of results.
4) In line 139-140, authors need to add more information. Increase (141.24% and 74.23%) of soluble proteins should be obtained from the condition of the treatment of NaCl “with MT pretreatment”. In addition, authors showed enhanced contents of soluble proteins and proline by NaCl and MT. If authors used the contents of soluble proteins and proline as osmotic adjustment, why those were not increased in the condition of the treatment of NaCl without MT?
5) In line 195-205, authors suggest the possible role of MAPK cascade, which is induced by H2O2 pathway. Although authors showed induced expression of MAPK cascade genes, it does not show the direct correlation with the involvement of MAPK cascade. To suggest this hypothesis, authors need to try to show the MAPK activation by salt stress. Activated (phosphorylated) MAPKs could be detected by an anti-phospho-Erk1/2 antibody (Cell signaling) in an immune blot analysis.
Author Response
Dear reviewer,
Thank you very much for your advice. We have revised the manuscript, addressed the comments, and marked the amendments in red in the revised manuscript. Point by point responses to the your comments are listed below this letter. The manuscript that we have submitted had been polished in the language by a native English speaker. If you have any questions, please don’t hesitate to contact me at any time.
Thank you and best regards,
Yours sincerely,
Yingjuan Wang
Replies to Reviewer :
Major concerns
1. Most of all, my major concern in this manuscript is that authors used low number of samples in all physiological results. These low number of samples did not seem to show statistical meaning. As shown in Figure 1, I could find several number of leaves even in one seedling. What does 'n=3' mean? Authors must clarify the number of samples, how you selected samples and how you used them in measurement in figure legend or materials and methods. Or authors seem to need to use an increased number of samples in physiological experiments.
Answer:Thank you for your kindly suggestions. This may be a misunderstanding caused by not rigorous description. 'n=3' indicates that the biological and experimental replicates were three times.We have charified the number of samples, how you selected samples and how you used them in measurement in materials and methods in the revised version (lines 388-393). Charified the sentence: In the measurement process, each treatment group contained 90 strains seedlings (each contained 30 strains in three parallel control groups), and randomly selected six strains of 90 seedlings in three replicates for various indicators, and each index was repeated in the biological and experimental for three times (that is, in the biological and experimental replicates for three times, 18 strains seedlings were randomly selected from 270 seedlings).
Minor concerns
2. In line 51 and 54, authors need to correct gene names, such as MAPK6 (line 51) and MAK6 (line 54), as “MPK6” based on references what authors provided.
Answer: Many thanks for your helpful suggestions. We have corrected MAPK6 (line 51) and MAK6 (line 54) as “MPK6”( lines 51, 54)in the revised manuscript.
3. In line 101, authors need to change unclear sentence, “After 6 days of the salt plus MT pretreatment” like as “Measurement was carried out after 6 days of the salt plus MT pretreatment” in each figure legend.
Answer: We have changed sentence “After 6 days of the salt plus MT pretreatment”into “Measurement was carried out after six days of the salt plus MT pretreatmemt”in figure legends 2-9 in the revised version (lines 113, 130-131, 150-151, 171,190-191, 212, 221-222, and 259-260).
4. In line 126-128, authors described the conclusion of the biological role of MT to enhance antioxidant enzyme activities. However, authors showed the attenuated result of SOD in Figure 3A, whereas other enzymes displayed the opposite roles in the condition containing MT. Is there any different effect of SOD compared to other antioxidant enzymes including POD, CAT and APX? Authors need to add more discussion for this or change the conclusion of results.
Answer: We are very grateful to you for your constructive and helpful comments. In the study, after treatment with MT alone, the activities of various antioxidant enzymes (SOD, POD, CAT, and APX) increased (Figure 3), and there was no " showed the attenuation of SOD, whereas other enzymes contained the opposite roles in the condition containing MT ". After MT treatment, the activity of SOD was most affected by MT. After salt stress, the activities of various antioxidant enzymes (SOD, POD, CAT and APX) also increased, and the activities of various antioxidant enzymes (SOD, POD, CAT and APX) were more increased after salt stress with MT ptetreatment (Figure 3). The SOD activity was only slightly weaker than that of the melatonin-treated group alone (Figure 3A).
5. In line 139-140, authors need to add more information. Increase (141.24% and 74.23%) of soluble proteins should be obtained from the condition of the treatment of NaCl “with MT pretreatment”. In addition, authors showed enhanced contents of soluble proteins and proline by NaCl and MT. If authors used the contents of soluble proteins and proline as osmotic adjustment, why those were not increased in the condition of the treatment of NaCl without MT?
Answer: Thank you for your kindly suggestions. In our study, the proline content increased (Figure 5B) but the soluble protein content decreased (Figure 5A) under salt stress compared to the control group. Reasons for the decrease of soluble protein content: under salt stress, hypoxia causes a large amount of peroxidation products in plants, causing damage to soluble proteins, resulting in a decrease in content [1].
1. Wang, J.Z. Changes in photosynthetic properties and antioxidative system of pear leaves to boron toxicity. Afr. J. Biotechnol. 2011, 10, 19693–19700. [CrossRef]
6. In line 195-205, authors suggest the possible role of MAPK cascade, which is induced by H2O2 pathway. Although authors showed induced expression of MAPK cascade genes, it does not show the direct correlation with the involvement of MAPK cascade. To suggest this hypothesis, authors need to try to show the MAPK activation by salt stress. Activated (phosphorylated) MAPKs could be detected by an anti-phospho-Erk1/2 antibody (Cell signaling) in an immune blot analysis.
Answer:Thank you for giving us this good advice and also giving us a good idea. We agree to use western blot analysis to directly verify the direct correlation between the H2O2signaling pathway and the MAPK cascade. In the present study, the results could not confirm the direct correlation with the MAPK cascade. Therefore, according to the results of the study, the presence of MT might induce the expression of MAPK cascade gene and we only speculated that MT might participate in the induction of MAPK cascade through H2O2 pathway.
Once again, we would like to express our sincere thanks to the reviewer for the constructive and positive comments.

Reviewer 3 Report
In this paper the Authors reported the melatonin-mediated regulation of growth and antioxidant capacity in salt-tolerant naked oat under salt stress. The topic is interesting and fits the overall scope of the journal. Novel and useful findings have been reported. Moreover, the paper is well organized and the data properly presented and discussed. In my opinion, the paper merits the acceptance after minor revision as I reported in the attached full-text PDF file.

Author Response
Dear reviewer,
Thank you very much for your advice. We have revised the manuscript, addressed the comments, and marked the amendments in red in the revised manuscript. Point by point responses to the your comments are listed below this letter. The manuscript that we have submitted had been polished in the language by a native English speaker. If you have any questions, please don’t hesitate to contact me at any time.
Thank you and best regards,
Yours sincerely,
Yingjuan Wang
Replies to Reviewer :
In this paper the Authors reported the melatonin-mediated regulation of growth and antioxidant capacity in salt-tolerant naked oat under salt stress. The topic is interesting and fits the overall scope of the journal. Novel and useful findings have been reported. Moreover, the paper is well organized and the data properly presented and discussed. In my opinion, the paper merits the acceptance after minor revision as I reported in the attached full-text PDF file.
1. Line35, please add also this references:
Cristiano, G., Camposeo, S., Fracchiolla, M., Vivaldi, G. A., De Lucia, B., & Cazzato, E. (2016). Salinity differentially affects growth and ecophysiology of two mastic tree (Pistacia lentiscus L.) accessions. Forests, 7(8), 156.
Answer: Cristiano et al. suggests that salinity affects plant growth. This reference has been quoted as ref. 2 in the revised version (lines 490-492).
2. Line279, please delete P value
Answer: Many thanks for your constructive comments. We have deleted P values (p < 0.01) in the revised version (line 311).
Once again, we would like to express our sincere thanks to the reviewer for the constructive and positive comments.

Round 2
Reviewer 2 Report
The authors addressed all my concerns adequately and, in my view, the manuscript is ready for publication.
Author Response
Dear reviewer,
Thank you very much for your advice. We have revised the manuscript, addressed the comments, and marked the amendments in red in the revised manuscript. Point by point responses to the your comments are listed below this letter. The manuscript that we have submitted had been polished in the language by a native English speaker. If you have any questions, please don’t hesitate to contact me at any time.
Thank you and best regards,
Yours sincerely,
Yingjuan Wang
1. Please carefully check the super and subscripts of your abbreviations for ROS species so they match those used in the literature.
Answer: Many thanks for your constructive comments. We have rechecked the super and subscripts of your abbreviations for ROS species in the revised version (lines 42,175-176,179,190,285-286,445, 450-451,454,487).
2. Please adjust figure 5, 7 and 8 so the font sizes are all the same
Answer: We are very grateful to you for your constructive and helpful comments. We have adjusted the font sizes in figure 5, 7 and 8 (lines 166,209,221) and adjusted the font sizes in figure 6 (line 187) in the revised manuscript.
Once again, we would like to express our sincere thanks to the reviewer for the constructive and positive comments.
